# Optimal Coordinated Control Strategy of Clustered DC Microgrids under Load-Generation Uncertainties Based on GWO

Zaid Hamid Abdulabbas Al-Tameemi [1] , Tek Tjing Lie [1,*] , Gilbert Foo [1] and Frede Blaabjerg [2]

1 Department of Electrical and Electronic Engineering, Auckland University of Technology, Auckland 1010, New Zealand; zaid.altameemi@autuni.ac.nz (Z.H.A.A.-T.); gilbert.foo@aut.ac.nz (G.F.)
2 Department of Energy Technology, Aalborg University, 9220 Aalborg, Denmark; fbl@et.aau.dk
* Correspondence: tek.lie@aut.ac.nz

**Abstract:** The coordination of clustered microgrids (MGs) needs to be achieved in a seamless manner to tackle generation-load mismatch among MGs. A hierarchical control strategy based on PI controllers for local and global layers has been proposed in the literature to coordinate DC MGs in a cluster. However, this control strategy may not be able to resist significant load disturbances and unexpected generated powers due to the sporadic nature of the renewable energy resources. These issues are inevitable because both layers are highly dependent on PI controllers who cannot fully overcome the abovementioned obstacles. Therefore, Grey Wolf Optimizer (GWO) is proposed to enhance the performance of the global layer by optimizing its PI controller parameters. The simulation studies were conducted using the well-established MATLAB Simulink, and the results reveal that the optimized global layer performs better than the conventional ones. It is noticed that not only accurate power-sharing and proper voltage regulation within ±1% along with fewer power losses are achieved by adopting the modified consensus algorithm for the clustered DC MGs, but also the settling time and overshoot/undershoot are reduced even with the enormous load and generation changes which indicates the effectiveness of the proposed method used in the paper.

**Keywords:** DC microgrid; voltage regulation; power sharing; hierarchical control strategy; GWO

## 1. Introduction

A microgrid (MG) is a collection of localized distributed generator units (DGs), energy storage devices, and localized loads that function together in small-scale power systems [1]. This concept has been presented as a realistic alternative for integrating various renewable energy sources to construct a local grid [2]. Several types of MGs have been stated in the literature, including alternating current microgrids (ACMGs), direct current microgrids (DCMGs), and hybrid AC/DC MGs. This categorization is based on the type of connection bus employed in the system.

The DCMG is the focus of this study since it has advantages compared to ACMG, such as reliability, efficiency, and a simple control topology. Moreover, it does not require reactive power management and frequency synchronization [3]. These characteristics are the key reasons for DCMGs' extensive use as a principal source of power to fulfil rising demand in remote and small areas throughout the world. The DCMG can work in both autonomous and grid-interconnected modes, just like ACMG [4]. To increase the flexibility and dependability of the DCMG, different MGs near each other may be joined to form an MG cluster, which may then be coupled to the utility grid [5]. The following are the significant advantages of an MG cluster: (i) boosts renewable energy penetration while also expanding the power supply region, (ii) improves the MG's stability and dependability in tackling generation uncertainties and demand fluctuations; and (iii) enhances operational efficiency, flexibility, and economy of the whole system [5,6].

Such MG clusters need to be coordinated seamlessly to handle generation-load mismatches and maintain the voltage regulation of participating MGs within a particular limit. The literature review found that different control strategies have been used to coordinate DCMGs in a cluster, including centralized, decentralized, distributed, and hierarchical control strategies [7]. Centralized control schemes are those in which the central controller connects with all the units over bidirectional high-bandwidth communication links to acquire the information necessary to formulate the command signals. These control techniques have been introduced in [8–10] for the purpose of managing the interconnection of DCMGs in order to ensure adequate and well-organized functioning. Although it demonstrates efficient and precise functioning, it is susceptible to a single point of failure (SPOF); this jeopardizes the system's stability in addition to predisposing it to cascade failures and ultimately collapse [11] as well as it necessitates the use of complicated communications networks, which reduces the entire system's reliability and expandability [12,13]. In general, it has been observed that centralized methods are more suited for regional and comparatively tiny MGs, where the amount of data to be gathered is limited [11,13]. Without recourse to the centralized control approach, the decentralized control method can deliver commands/orders to the local control layers in response to inputs from the converter or adjacent converters [14]. While the techniques do not entail a sophisticated communications network or a centralized controller, they lack the capability to supervise each converter appropriately. In this regard, distributed control techniques can be used to address the high bandwidth communication needs of the centralized control. As opposed to this, a distributed control strategy offers supervised control via a bidirectional low-bandwidth communication network to share data between neighbouring units, as well as reliable voltage adjustment and proportionate current distribution by the local controllers [15–17], so it is commonly utilized in managing MG clusters. It is noticed that centralized, decentralized, distributed control techniques do not have the ability to face the problems of the modern power system that become more complex, and its demand is unpredictable, mainly when the system incorporates a complicated decision-making mechanism [11]. In this respect, a hierarchical control strategy characterized by high control reliability and smooth operation of the MGs in a cluster is a proper control technique to be adopted to address such issues. This strategy may be divided into three control layers: primary, secondary, and tertiary, as explained in Section 3.

This concept has been adopted to manage the clustered MGs based on the consensus algorithm that is generally used in [1,5,6,14,18–21]. A distributed hierarchical control approach has been presented in [20] to ensure DC clusters MGs in the system operate efficiently and reliably. While both voltage regulation and power transfer management can be achieved using this technique efficiently, this approach may not always be reliable for real application scenarios because it does not consider power losses and the fulfilment of balanced regulated SOCs. Using SOC-featured distributed tertiary control (DTC), reported in [22], all ESSs in a cluster can be coordinated, and power allocated automatically based on the levels of each MG's ESS. It is worth mentioning that during the process of charging and discharging in the simulated MGs cluster, the control approach can immediately bring the SOC and output current of each EES to consensus. However, it is mainly vulnerable to any emergency circumstance because its dependency on the ESS adjustment factor affects its convergence. One method of jointly controlling several MGs in a cluster is described in [23] as a distributed tertiary control system that modifies each MG's voltage setpoint depending on its connected load. Despite the fact that this approach was shown to boost MG load distribution and fault resiliency, the unpredicted difference in grid voltage, which this study does not address, can cause a current to flow even without the necessity for power distribution. In [14], a two-level distributed control strategy was proposed for an MGC and was shown to be able to manage DC-link voltage with precise current distribution amongst multiple converters. The main issue of this method is that a complicated communications system is required to deliver an appropriate signal reference from the global control level to the primary level, increasing the costs of communication topology. In addition, RES/load

inconsistency and fault circumstances have not been explored to evaluate the suggested method's efficacy in real-world scenarios, and this is a significant shortcoming of the approach. A few of the concerns raised in [14] have been addressed by the authors of [21], who propose an alternative two-level tertiary control approach that uses pinning to modify the setpoint voltage of each MG and balance the loads across all connected sources in the MGC. As a result, the system seems to be more secure, more reliable, and less vulnerable to physical and cyber threats owing to this new technique. However, diverse sources of energy are ignored in favour of AC-to-DC conversions via buck converters. This method also fails to account for the inherent uncertainty of RESs (PV and WT). Accordingly, the suggested control approaches of the foregoing research have not been tested with regard to uncertainty in the RES of their DGs. Therefore, in [5], it is proposed to use a MAS-based coordinated power regulation technique with virtual inertia (VI) in DC MG clusters. This technique was put through its paces under the influence of load-RES uncertainty to test the simulated DC cluster's ability to provide accurate power sharing and global voltage regulation. Many difficulties have been brought out in prior references, excluding [5], such as the lack of consideration for the variety of energy sources and the inherent uncertainty of RESs (PV AND WT) and realistic load variations. Additionally, it was discovered that these articles provide a distributed control mechanism based on the linear consensus notion, which exposes MGs to rapid interruptions caused by intermittent renewable energy sources (PV and WT) and unexpected load fluctuations.

Although many developments have been undertaken to boost the performance of the hierarchy control approach, it is tough to project a proper consensus protocol that can achieve an optimum power exchange among coupled MGs because it is typically reliant on PI controllers in the global control layer of the hierarchical control scheme. Although these controllers are featured by their simple design and reasonable performance, they might not be effective for non-linear processes under various working conditions. Thus, a robust method to achieve a suitable tuning may be required.

In this regard, the outstanding role of meta-heuristic optimization approaches in addressing complex problems and optimizing controller parameters more competently [24] can be exploited to tackle the main issues in the global control layer of clustered DC MGs, which are identified in the previous studies. Therefore, a grey wolf optimizer (GWO) which is featured by quick convergence, simplicity, ease of implementation [25], and superior performance in unknown, difficult search spaces, especially in engineering applications [26], has been demonstrated to outperform the PSO algorithm, bat algorithm, ant lion algorithm, and gravitational search algorithm [27]. Therefore, it is utilised in this article to modify the main issues of the consensus algorithm that is used in the global layer to face load-generation uncertainty which normally occurs in a real-life scenario. In other words, a modified consensus algorithm is adopted for the first time to coordinate DCMGs properly in the simulated cluster.

In summary, the main contribution of this paper is to utilize a meta-heuristic optimization algorithm, in this case, GWO, in the top layer of the hierarchical control scheme. The proposed control scheme is shown to be able to keep the DCMGs cluster operating steadily (it closely follows the reference signals) under critical operating conditions. In addition, it can improve efficiency by reducing the power losses in the cluster.

## 2. Configuration and Control

A typical clustered MG incorporates several forms of distributed energy resources such as a photovoltaic (PV), wind turbine generation, and fuel cell accompanied by energy storage systems (ESSs) to tackle the dilemma of such resources' sporadic availability [1]. The majority of such resources produce DC electric power, with the exception of wind turbines, which generate AC power and may be integrated into the DCMG if converted [28]. To make the system more reliable, interconnected MGs are adopted to support each other in emergency cases [5]. For this study, the cluster comprises two MGs linked together using a resistive-inductive line as a tie-line, as evidenced in Figure 1. The primary sources in

both MGs include a PV array, battery, and a supercapacitor (SC) with local loads which are coupled to a DC-link using DC-DC converters. The clustered MGs may be able to work in two distinct modes: (i) island mode and (ii) interlinked mode.

In the island mode, each MG uses its own local control system to deliver reliable power to a connected regional load [11]. In the interlinked mode, the connected DCMGs utilize a coordinated power approach to provide power for the DCMGs cluster [29]. In addition, each sub-MG is connected to its neighboring MGs by communication lines, allowing for the flow of information between them. Each MG is designated an agent at this level of abstraction. Moreover, each MG gets information from the MGs that link them, consisting of an average voltage besides each MG's state of charge (SOC). The reference control signals will be delivered to the local layers of the clustered MGs based on a modified consensus algorithm to ensure optimal power-sharing among them and global voltage regulation [30].

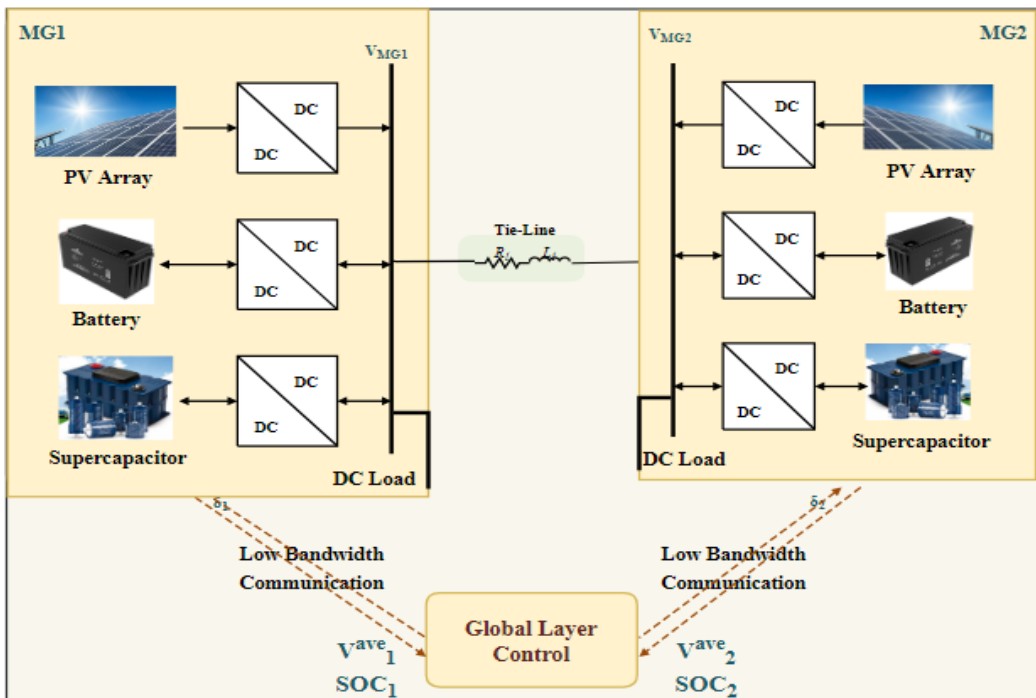

**Figure 1.** Interconnected DC microgrids configuration.

## 3. Hierarchical Control Strategy

Several existing research works have recommended the hierarchical control paradigm to handle control problems that come from incorporating DGs into an MG and the coordination of MGs in a cluster [31]. This technique comprises numerous control levels, enhancing the MG's flexibility and efficacy. The key benefits of this technique are the ability to classify the MG control system into several layers to ensure high control dependability and efficient functioning in the grid-connected besides independent modes [32]. The first level of the hierarchy scheme is in charge of the preliminary power-exchange scheduling and current /voltage regulation amongst the participant converters in each MG [33]. The secondary control level, which is a higher degree of control than the primary control, deals with voltage restoration and performance enhancement. At the apex of the hierarchical system, the tertiary level is accountable for power management, energy management, system optimization, and economic dispatch [28]. This method is introduced to manage the clustered MGs based on the consensus algorithm extensively employed in the literature. All these control levels are explained in Figure 2.

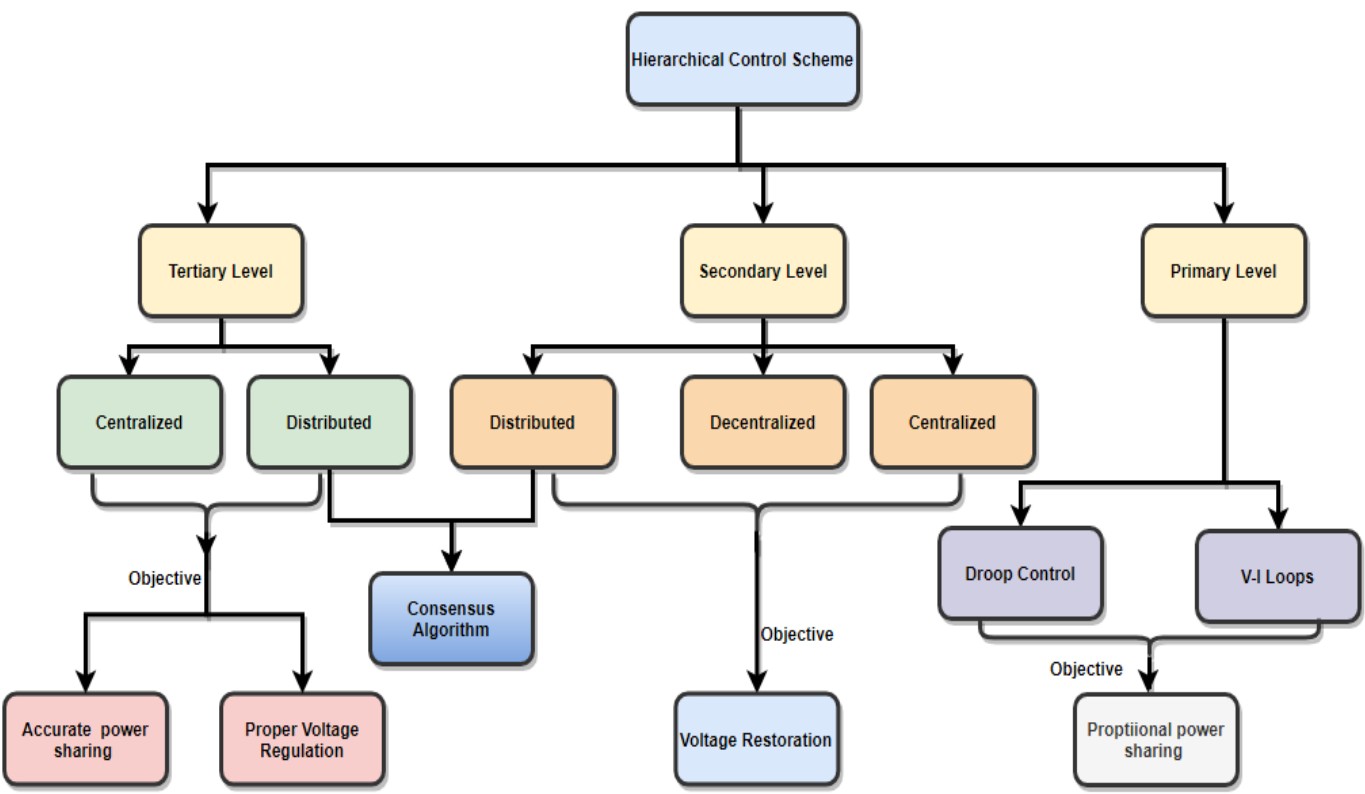

**Figure 2.** Hierarchy Control Strategies.

### 3.1. Local Control Layer

The primary control level, which is identified as the local control layer, can be utilized to coordinate DGs within an MG [28]. The chief goal of this control layer is to manage the DC-DC converters' output current and voltage in each MG. Various control techniques, including droop control, DC bus signaling, and fuzzy-logic controller, have been used in the literature to be applied to this layer, which are based entirely on utilizing the local information of an MG in addition to keeping voltage stability. However, droop control with current and voltage loops (V-I) is prevalently used in this level [17]. This is due to the fact that it facilitates power transfer with no need for communication amongst the connected sources inside the MG, which is recognized as one of its most important advantages. Thus, there are no concerns about faults in communication because of this reason [32]. However, the trade-off between power exchange and voltage adjustment/regulation is a drawback of this technology [34]. The primary controllers may take on several forms based on various input sources for the converter module, such as charge/discharge control for ESS and maximum power point tracking (MPPT) for both PV panels and wind turbines [29]. This study uses the control strategy in [25] to construct the local control layer. It is also improved it by limiting high-frequency components of current that pass to the SC, which have not been considered in [35]. This contributed to reducing an unwanted power exchange; hence lower tie-line power losses are achieved. Due to its dependency on the local variables only, the essential performance of the MG may not be realized. Therefore, the top-level controllers are required to communicate with the local layer by utilizing their local variables to accomplish the desired performance [11]. It performs proper control actions over converters whenever it acquires the setpoints provided by upper layer controllers, as shown in Figure 3. Therefore, this paper focuses on the global control layer, which plays a vital role in addressing all drawbacks of the control method used in the local control layer. In the following subsection, the global control layer will be thoroughly developed.

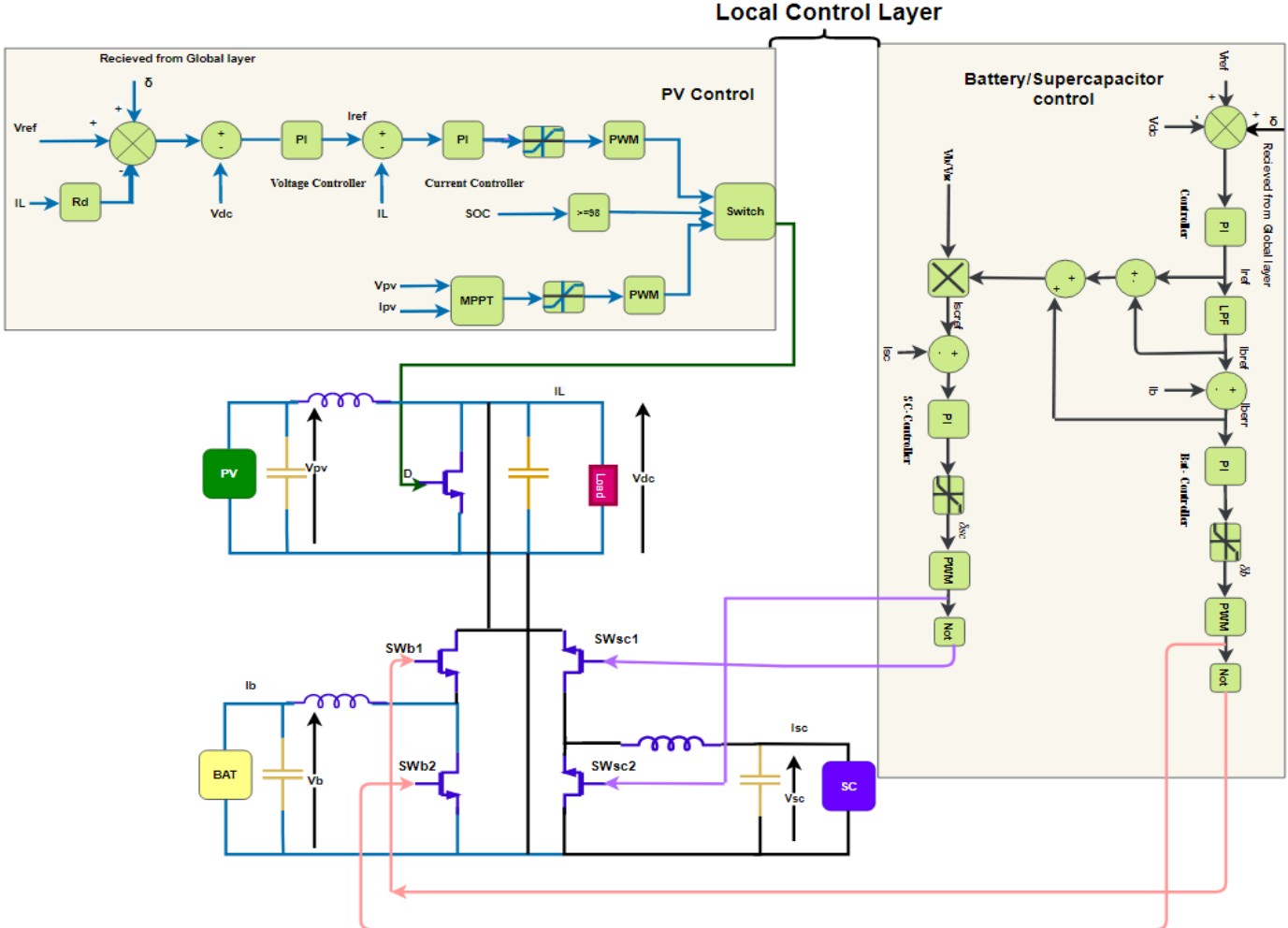

**Figure 3.** Local control layer.

*3.2. Global Control Layer*

3.2.1. Secondary Control

The secondary control is the second level of the hierarchy control scheme that can be conducted by using a centralized, decentralized, or distributed control strategy depending on communication linkages among agents in the system [17]. It is important to mention that it may not be proper to implement it using a centralized or decentralized approach due to their drawbacks as discussed in Section 1, which reduce the reliability of the system in providing reliable and stable power to consumers under critical operating conditions. Therefore, a distributed control strategy can be adopted at the secondary control level to achieve the main objective of this level. The main feature of adopting this strategy is that it enables the system to preserve complete functionality even if some lines of the communications networks fail [28]. As a result, distributed control is resistant to the effects of an SPOF. To be clear, regardless of their differences in design, their goal is the same: regaining or eliminating the voltage variation caused by the primary control. However, a digital communication connection (DCC) is needed when controlling both centralized and distributed systems. Therefore, the reliability of both control techniques will be reduced. In this context, a distributed secondary voltage control scheme based on a dynamic consensus mechanism is selected to be adopted [19]. It is essential to mention that the development of a distributed controller for interlinked DCMGs that may operate in off-grid or grid-connected configurations has made significant progress. To realize the key objectives, distributed secondary controllers are required to accomplish both effective voltage adjustment and proportional load

allocation amongst the local DGs by sending voltage correction $\delta_{Vr}$ to the local control layer while remaining resilient to communication connection unpredictability as well as cyber-attacks and other threats [12]. In this method, the distributed protocol at each agent is formulated in Equations (1) and (2) as follows:

$$V_i^{ave} = \sum_{j \ni Ni} a_{ij} \left( V_j^{ave} - V_i^{ave} \right) + V_i \tag{1}$$

$$\delta_{Vr} = \left( Vref - V_i^{ave} \right) \cdot \left( K_{pv} + S^{-1}K_{iv} \right) \tag{2}$$

where i and j represent MG1 and MG2 while $V_i^{ave}$ and $V_j^{ave}$ refer to the estimated bus voltages at MG1 and MG2, respectively. Whereas the adjacency matrix of communication topology among MGs and measured voltage of MG1 are denoted as $a_{ij}$ and $V_i$, respectively. It is worth mentioning that $a_{ij} = 1$ means both MGi and MGj are linked; otherwise, there are not interrelated, as shown in Figure 4. Additionally, $\delta_{Vr}$ and Vref embody voltage compensation and the reference voltage of MG1 and MG2 in a cluster, respectively.

### 3.2.2. Tertiary Control Level

The tertiary control is the third part of a hierarchy control approach, and it is responsible for handling the power distribution amongst MGs, and an external electric grid, which might be the grid or perhaps another MG [12,36]. It is worth mentioning that the hierarchical control scheme's most sophisticated level is this one among the other control levels. This controller has been a vital tool for power and energy management with the advent of MG networks. Although the MG is much smaller than a regular grid, the need to manage the flow of power and energy management is vital to improving the system's overall efficiency. Third-level control can be achieved in a hierarchical control scheme, either centralized or distributed, depending on the situation [12]. Because of the main issues of the centralized control strategy mentioned in Section 1, the distributed control system appears to be a viable alternative to the centralized control system since it provides greater reliability and scalability while utilizing an uncomplicated communication network. The communication network is designed in such a way that each unit communicates data only with the two units that are closest to it [12,36]. As a result, the distributed power management has been the focus of the DCMGs' research-related field. The principal purpose of this level is to realize optimal energy scheduling, energy storage, and power-exchange adjustment [11]. Using the simple tuning techniques of controllers may make the tertiary controllers slow and useless. It is worth mentioning that the distributed protocol of this control level at each agent/MG is expressed in Equations (3) and (4) as follows:

$$SOC_i^{ave} = \sum_{j \ni Ni} b_{ij} \left( SOC_j^{ave} - SOC_i^{ave} \right) + SOC_i \tag{3}$$

$$\delta_{Pf} = \left( SOC_i^{ave} - SOC_i^{ave} \right) \cdot \left( Kp_P + S^{-1}(Ki_P) \right) \tag{4}$$

The estimated SOCs of both batteries in MG1 and MG2 with their measured values are symbolized by $SOC_i^{ave}$ and $SOC_i$, respectively. Power compensation and adjacent matrix, which are shown in Figure 4, are signified by $\delta_{Pf}$ and $b_{ij}$, respectively. Both $\delta_{Pf}$ and $\delta_{Vr}$ are summed to be sent as $\delta$ to the local control layers in each MG (as shown in Figure 3) to restore normal DC-bus voltage and achieve proper power allocation amongst DGs in each MG and also between participating MGs.

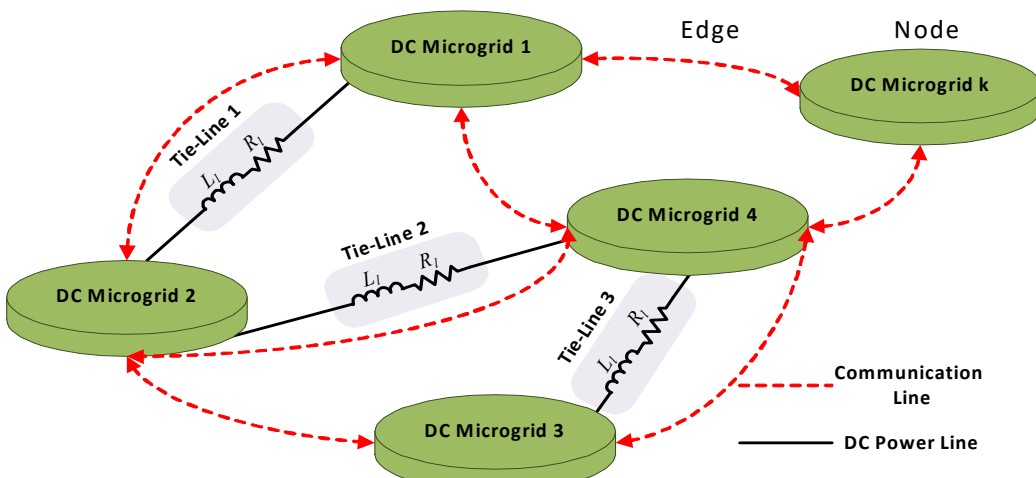

**Figure 4.** Communication Topology of DC MGs cluster [7].

## 4. Proposed Control Technique

For this study, the GWO, which is characterized by quick convergence, simplicity, and ease of implementation [25], is adopted to optimize PI controller parameters used in the global control layer, as revealed in Figure 5. The proposed technique can boost the power exchange and the global voltage of the clustered DCMGs. The proposed fitness function is expressed in Equation (5) as follows:

$$\text{ITAE} = \int_0^\infty t|\text{Er(t)}|\text{dt} \quad \text{Erp} = \text{SOC}_i - \text{SOC}_i^{\text{ave}} \quad \text{Erv} = V_{\text{dc}}^{\text{ref}} - V_i^{\text{ave}} \tag{5}$$

where ITAE—integral of time-weighted absolute error, Er(t)—the difference between the setpoint and the controlled variable, and t—time. It is worth mentioning that Erp and Erp refer to errors of the power regulator and voltage regulator, respectively. The reduction of the ITAE is frequently employed as a tuning criterion to obtain controller PI parameters [37]. The proposed control approach is explained in detail in Section 4.2.

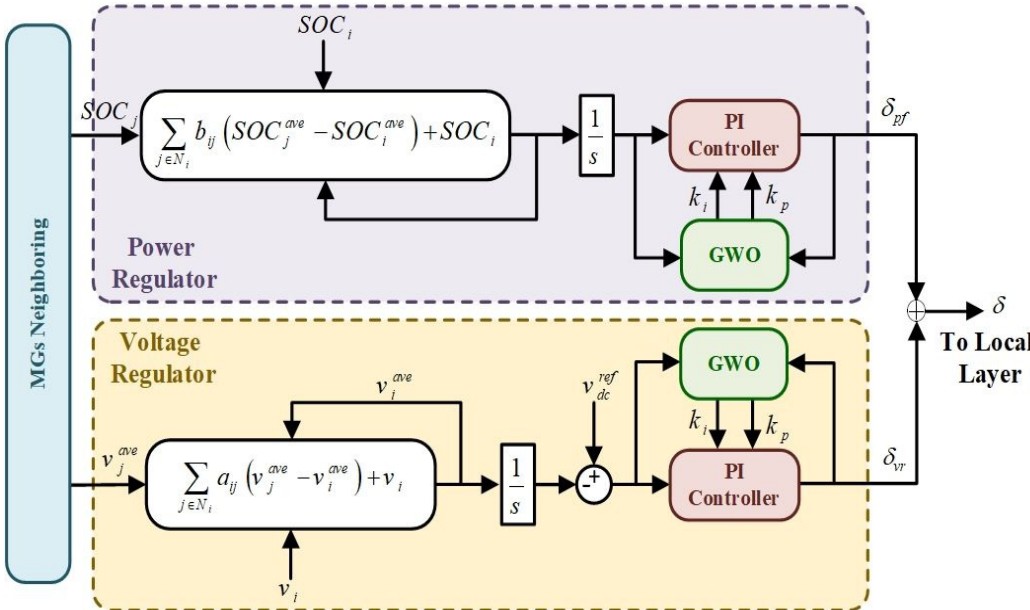

**Figure 5.** Proposed Control Method.

### 4.1. Grey Wolf Optimizer

Grey wolf optimizer (GWO) is one of the metaheuristics algorithms that has been proposed by [26] to simulate the social hierarchy and hunting mechanism of grey wolves in nature [38]. This algorithm is built based on three basic steps: seeking prey, surrounding prey, and launching attacks to achieve the required mission. To represent the wolf pack's leadership structure mathematically, consider alpha ($\alpha$) as the best option and beta ($\beta$) and delta ($\delta$) as the second and third best options, respectively, as depicted in Figure 6. Omega ($\omega$) is considered for the remaining possible solutions. It is worth mentioning that the best three wolves lead the hunting (optimization) in the GWO algorithm, while the rest of the wolves should update the place depending on the best three wolves' sites. Regarding the suggested algorithm, the equations of GWO can be stated in Equations (6) and (7) as follows:

$$\vec{D} = \vec{C} \cdot \vec{X_p}(t) - \vec{X}(t) \tag{6}$$

$$\vec{X}(t+1) = \vec{X_p}(t) - \vec{A} \cdot \vec{D} \tag{7}$$

where (t) represents the current iteration of the wolf/prey, $\vec{A}$ and $\vec{C}$ signify coefficient vectors as shown in Figure 7, $\vec{X_p}$ (t) indicates the vector of prey location, and $\vec{X}(t)$ means the vector of wolf placement.

The vectors $\vec{A}$ and $\vec{C}$ may therefore be expressed in Equations (8) and (9) as follows:

$$\vec{A} = 2\vec{a} \cdot \vec{r1} - \vec{a} \tag{8}$$

$$\vec{C} = 2 \cdot \vec{r2} \tag{9}$$

where r1 and r2 are included as arbitrary vectors in [0, 1], and components of $\vec{a}$ are lowered linearly from 2 to 0 over the iteration process. There are two randomized vectors, r1 and r2, allowing the wolves to go in any direction. A grey wolf can change its location in the area surrounding its prey at any time using Equations (8) and (9).

Prey can be identified and surrounded by grey wolves. Generally, the $\alpha$ is in charge of the chase. The $\beta$ and $\delta$ may be able to go hunting every now and again. However, no one has any notion where the optimal is situated in an arbitrary search space (prey). Grey wolves' hunting behavior may be represented mathematically by assuming the best candidate solution ($\alpha$) has better information about the possible locations of the targeted prey. The top possible solutions found thus far are retained, forcing the other wolves to change their existing positions to match the top search agents' sites—the subsequent Equations (10)–(16) are provided for this purpose.

$$\vec{D_\alpha} = \left| \vec{C_1} \vec{X_\alpha} - \vec{X} \right| \tag{10}$$

$$\vec{D_\beta} = \left| \vec{C_2} \vec{X_\beta} - \vec{X} \right| \tag{11}$$

$$\vec{D_\delta} = \left| \vec{C_3} \cdot \vec{X_\delta} - \vec{X} \right| \tag{12}$$

$$\vec{X_1} = \vec{X_\alpha} - \vec{A_1} \cdot \vec{D_\alpha} \tag{13}$$

$$\vec{X_2} = \vec{X_\beta} - \vec{A_2} \cdot \vec{D_\beta} \tag{14}$$

$$\vec{X_3} = \vec{X_\delta} - \vec{A_3} \cdot \vec{D_\delta} \tag{15}$$

$$\vec{X}(t+1) = \frac{\vec{X_1} + \vec{X_2} + \vec{X_3}}{3} \tag{16}$$

where the locations of the top three wolves such α, β, and δ, with respect to the targeted prey in the arbitrary search space, are denoted by $\vec{X_\alpha}$, $\vec{X_\beta}$ and $\vec{X_\delta}$, respectively. The placement of the present solution is symbolized by $\vec{X}$. Additionally, $\vec{X_1}$, $\vec{X_2}$ and $\vec{X_3}$ denote the ultimate locations of the top three wolves regarding the prey, respectively. Equations (10)–(12) are utilized to determine the distance between the current locations of these wolves and the prey. Equations (13)–(16) are used to calculate the ultimate site of the current solution depending on the calculated distance between the targeted prey and the corresponding wolves.

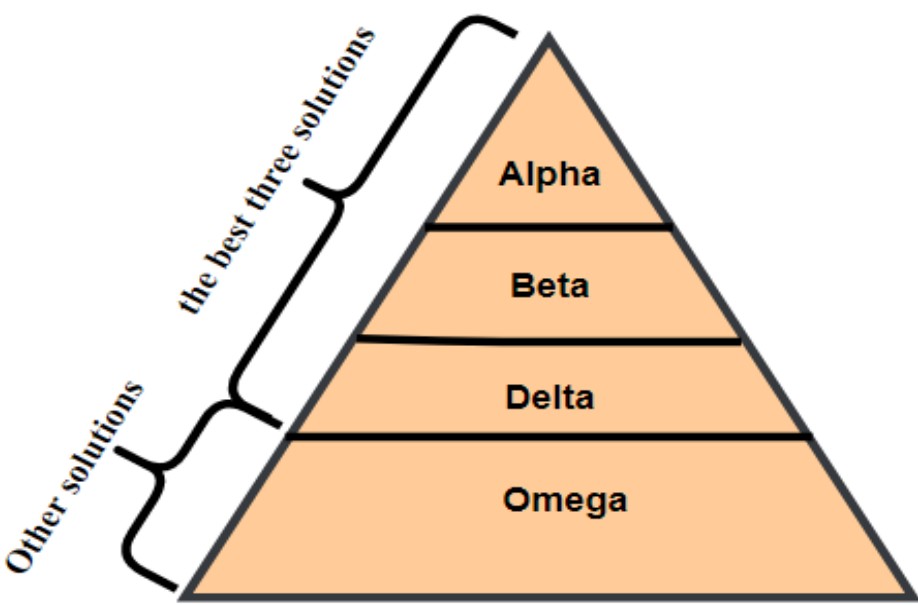

**Figure 6.** Hierarchy of grey wolves [26,38].

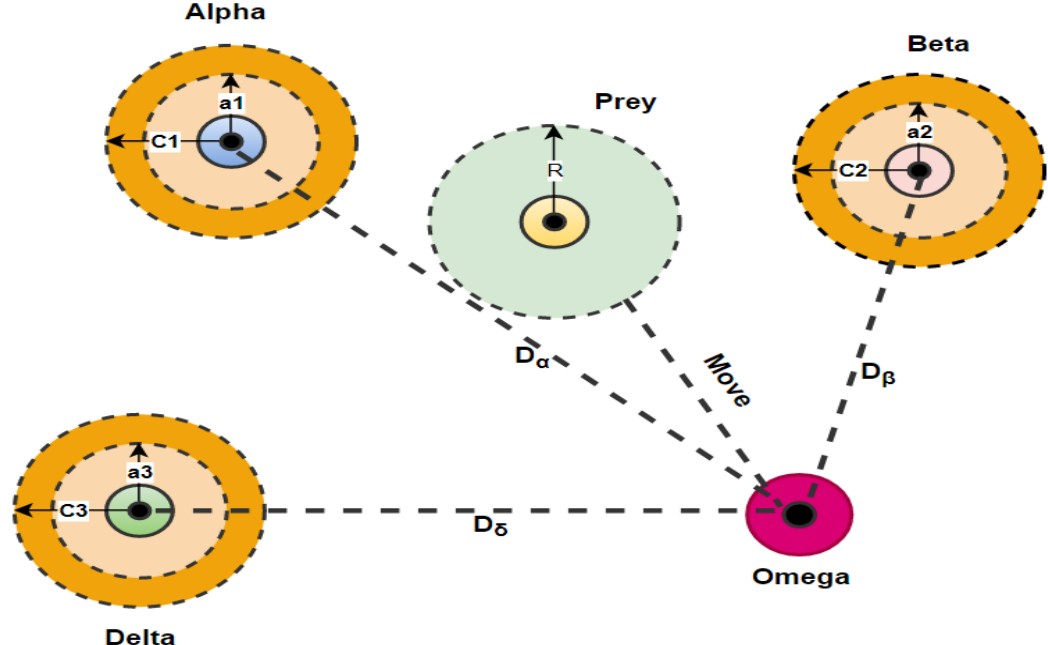

**Figure 7.** Positions updating in the Grey Wolf Optimizer (GWO) [26].

*4.2. Problem Formulation*

The key objective of this article is to improve the global layer of interconnected MGs based on the GWO technique, which is responsible for coordinating MGs in the cluster. By adopting this method, the controller parameters of the voltage compensator ($\delta_{Vr}$) and power regulator ($\delta_{Pf}$), which are formulated in Equations (2) and (4) are tuned effectively to enhance the performance of the DCMGs cluster in a matter of the power-sharing in addition to the voltage adjustment of DC-links. As stated in Equation (5), the fitness function (ITAE) is used to do this process. The ITAE of Equation (5) is optimized subject to the subsequent constraints as expressed in Equation (17).

$$\begin{cases} Kp_{min} \leq Kp \leq Kp_{max} \\ Ki_{min} \leq Ki \leq Ki_{max} \end{cases} \tag{17}$$

where $Kp_{min}$, $Kp_{max}$, $Ki_{min}$ and $Ki_{max}$ refer to the lower and upper limits of the PI-controllers in Equations (2) and (4).

To acquire the optimal values of Kp and Ki in this study, the following steps were taken:

i.   Generate the populations of grey wolves, $\overrightarrow{X_1}, \overrightarrow{X_2}, \overrightarrow{X_3}$ and so on, indicating that each grey wolf ($\overrightarrow{X}$) represents Kp and Ki in this study.

ii.  Initialize coefficients $\overrightarrow{A}, \overrightarrow{C}$ and $\overrightarrow{a}$ which are illustrated in Figure 7 so that their exploration and development capabilities may be used to improve the GWO algorithm's balance.

iii. Determine the top three wolves by calculating the fitness value of each agent (wolf).

iv.  In the case that the current iteration is less than the maximal iterations threshold, all other wolves ($\omega$) will update the places based on Step 3. Alternatively, the optimal values of X agents (Kp and Ki) will be attained to be implemented in Equations (2) and (4) to determine the best values of $\delta_{Vr}$ and $\delta_P$ for our system.

v.   Then all the coefficients $\overrightarrow{A}, \overrightarrow{C}$ and $\overrightarrow{a}$ will be updated in accordance with the first condition in Step 4. After that, the value of each search agent/wolf is re-calculated.

vi.  Based on the preceding updates, the best three wolves, $\alpha$, $\beta$, and $\delta$ are updated. This process is repeated until the best values of Kp and Ki are obtained.

It is worth mentioning that the steps (i–vi) are clearly depicted in Figure 8.

The steps mentioned above were implemented to get the best values of PI controllers based on the system requirements. Their parameters are obtained after running the MATLAB environment multiple times with the value of convergence as 0.006 as shown in Figure 9. It can be easily noticed that the GWO archives the minimum fitness value with a fewer number of iterations. It is worth mentioning the list of control parameters that are tuned by GWO are presented in Table 1.

**Table 1.** Optimal values of PI controllers.

| | |
|---|---|
| Kp1 | 8.3511 |
| Kp2 | 48.8918 |
| Ki1 | 3.0401 |
| Ki2 | 129.6347 |

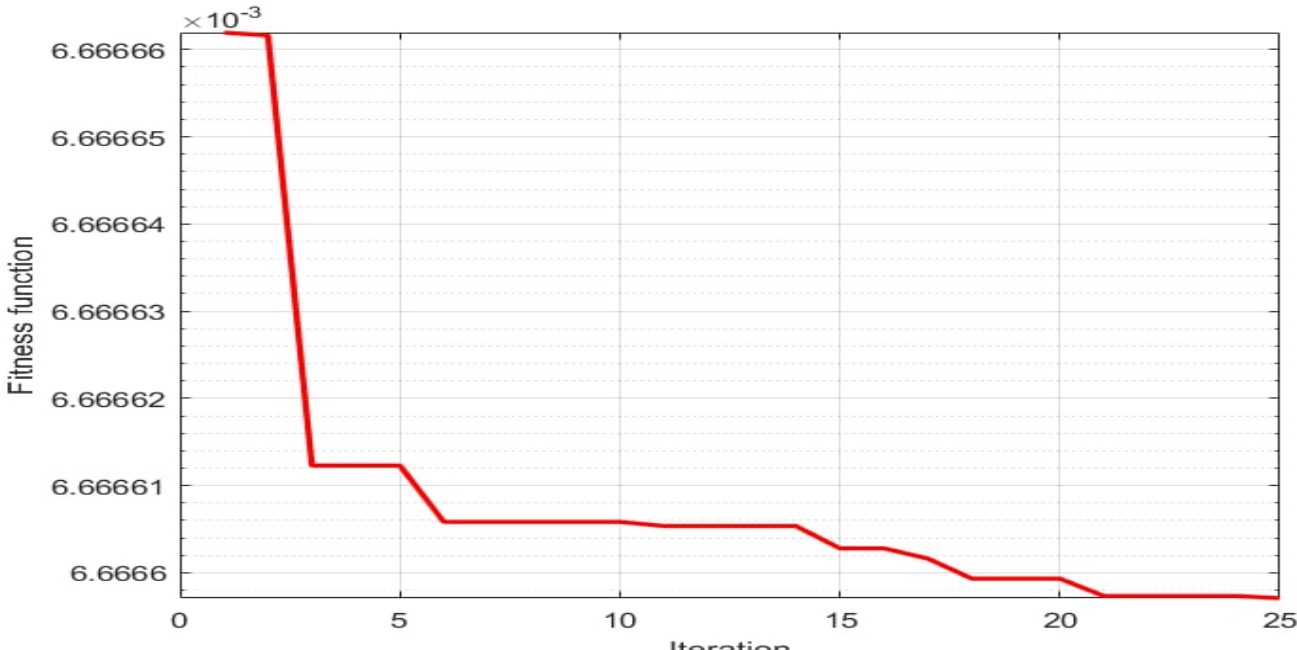

**Figure 8.** The GWO algorithm for the global control layer.

**Figure 9.** The convergence of GWO.

## 5. Results and Discussion

Figure 1 depicts the studied clustered MGs. Each MG has a PV array, hybrid battery/SC energy system, and DC load. It is essential to mention that the PV array has been set to 120.7W/panel (3 × 2 panels). A 24V, 14 Ah ESS, and 32V, 29F SC were chosen for the study. The initial load demand of both DCMGs was set to 300W with 48V as a reference voltage for both MGs. The components of each MG were interconnected to the DC bus by the mean of unidirectional and bidirectional converters. This study assumes that both energy storages have equivalent states of charges (SOCs). Different operating scenarios, including changes in the solar irradiance and load demand, are simulated to show the effectiveness of the proposed control scheme compared to that of the conventional consensus algorithm used in the top level of hierarchical control strategy (global layer).

These scenarios have been simulated with the existing consensus algorithm, and the results are presented in Figure 10. From Figure 10, one can see that load changes by the increase rate of 200 W and 100 W occur at MG1 at 2 s and 3 s, and MG2 at 2 s, as well as the solar irradiance which fluctuates at 2 s, 3 s, and 4 s, as demonstrated in Figures 10 and 11, respectively. From the reference values, the voltages drop to 46.61 V and 46.58 V at 2 s and 3 s. A maximum overshoot rises to 48.81V at 4s because of the solar irradiance variation during this period (see Figure 10a). In addition, it is found that the battery currents are not precisely tracking their reference values along with some unwanted batteries and SCs powers flow, and this is noticed in Figure 10b–e.

As demonstrated in Figure 11, these issues are addressed effectively with the proposed technique (optimized consensus algorithm). The obtained results in Figure 11a–e illustrate that although the studied clustered DC microgrid has been subjected to the critical operating conditions of both loads at 2 s, 3 s, and power generation variation at 4 s simultaneously, there are various advantages obtained by modifying the consensus algorithm, including maintaining the bus voltage within ±1% as illustrated in Figure 11a which indicates to the fast response of the proposed method, it is rejecting the imposed disturbances as compared to that of the existing control method. In addition, optimal power-sharing is achieved, as shown in Figure 11b, which eliminates the unwanted power-sharing among participating MGs in the cluster. Furthermore, Figure 11c shows that maximum battery currents provided to the cluster reduced from 7.5 A to 6 A during time intervals 3 s to 4 s because the load demand increases at both MGs to 500 W, which means less energy storage is required. It is essential to mention that battery currents in the cluster closely track their reference as opposed to the consensus algorithm demonstrated in Figure 10c. It is noticed that although PV generation increases at 4 s from 579 W to 688 W, the SOC of MG2 continuous discharging power to the cluster which indicates the low response of the consensus algorithm as compared to the performance of the proposed modified one, as shown in Figure 11d. Figure 11d explains the charging process of the SOC of MG1 at 4 s. In Figure 11e, transient perturbations have been successfully absorbed by the supercapacitors in both MGs, resulting in a significant reduction in disturbance rejection on the battery current, making it smoother. This indicates that this cluster becomes more reliable and steadier in providing continuous power to consumers under the abovementioned operating conditions.

It can also be noticed from Figures 10 and 11 that during start-up and load change conditions, overshoots/undershoots in the DC bus voltage, currents/voltages ripples, and settling time of interconnected DCMGs, the proposed method is better than the conventional consensus method, and this, in turn, can improve system reliability and decrease stress on the utilized components in the respective system. To fully comprehend the main features of the proposed control method compared with the conventional consensus algorithm widely used in the literature, the main results of both control algorithms are summarized in Figure 12.

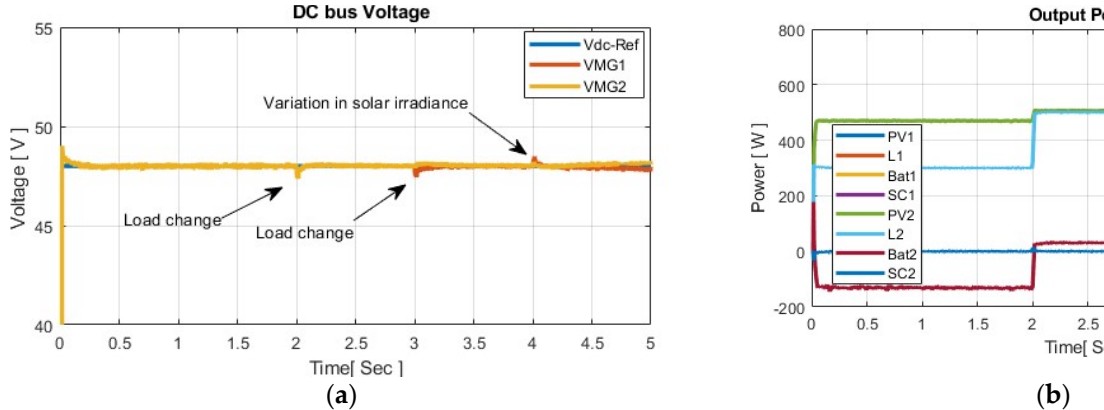

**Figure 10.** Results using consensus algorithm:(**a**) DC voltage; (**b**) output powers: (**c**) battery currents; (**d**) SOCs of batteries; (**e**) SOC of SCs.

**Figure 11.** *Cont.*

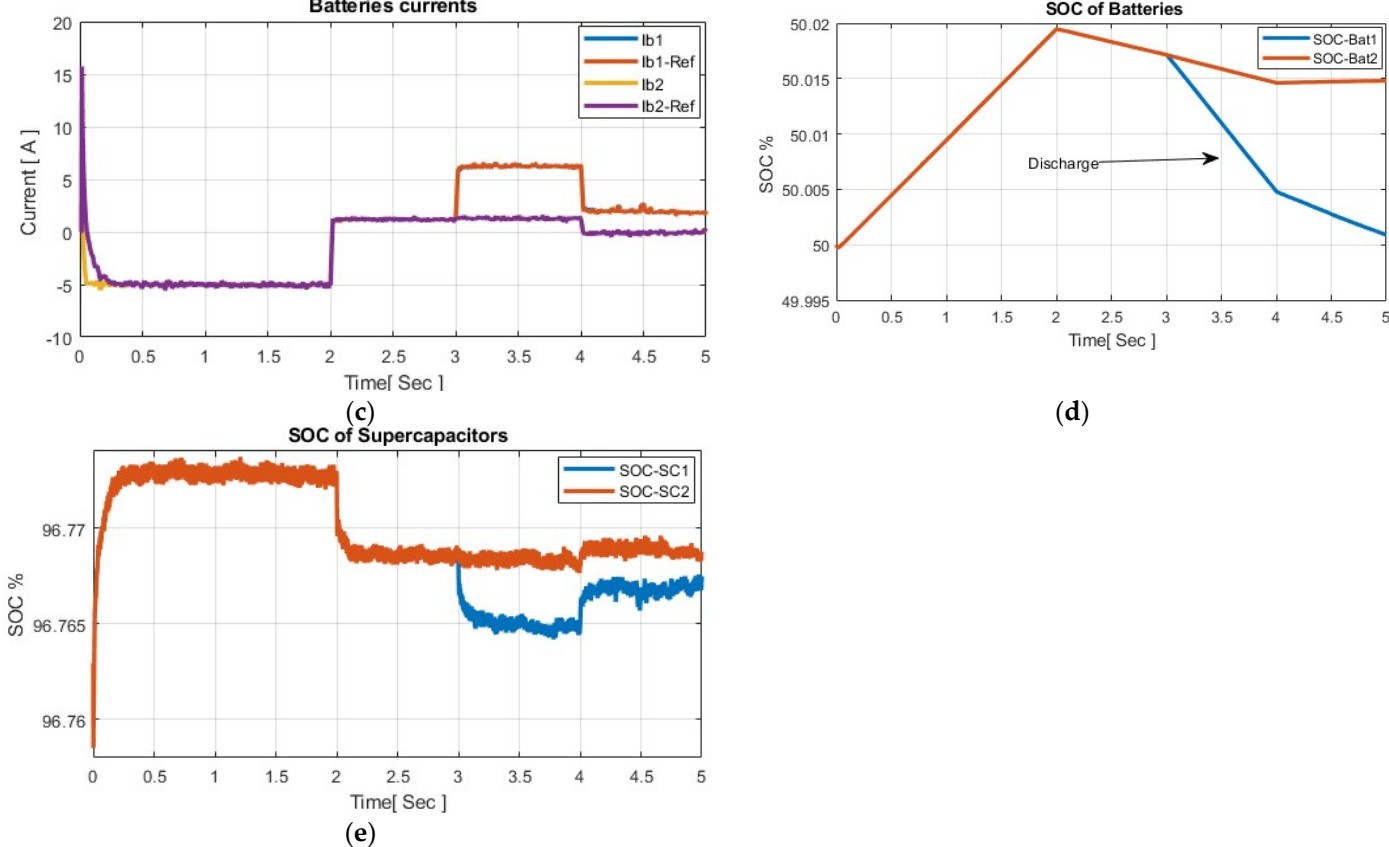

**Figure 11.** Results using the proposed method:(**a**) DC voltage; (**b**) output powers: (**c**) battery currents; (**d**) SOCs of batteries; (**e**) SOC of SCs.

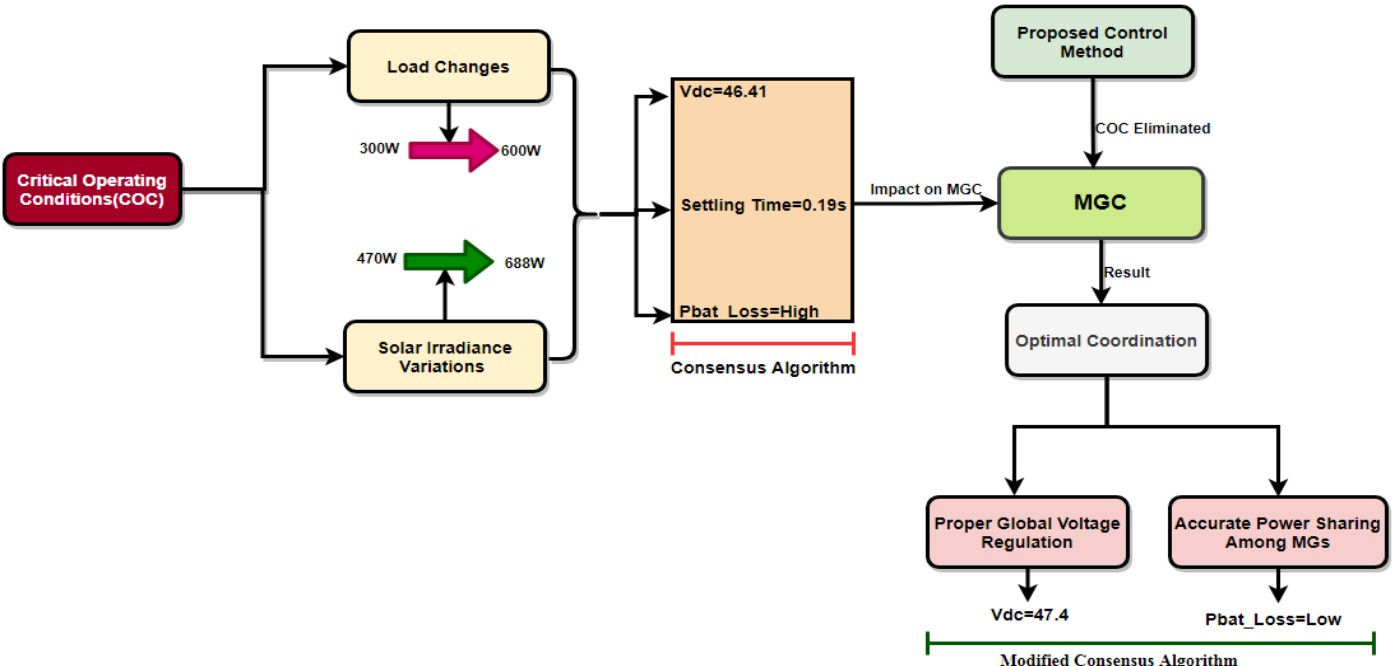

**Figure 12.** Summary of the main results.

**Remark 1.** *Based on the obtained results, this control method may be suitable for being implemented in real-life- scenarios as the noted issues in the previous articles, including voltage regulation and reduction of power losses, are considered to be solved in the paper.*

**Remark 2.** *It can contribute to reducing the dependency on classical grids by adopting the MGs cluster on a large scale to produce reliable and continuous power to consumers in/out remote areas under critical operating conditions.*

## 6. Conclusions

This paper proposed to utilise the GWO to enhance the global control layer of two DCMG clusters. The effectiveness of the proposed control scheme has been examined under several disturbances, such as changes in the solar irradiance and fluctuations in the load demand. The results showed that the proposed technique effectively adjusts the DC voltage and achieves precise load power-sharing among MGs, leading to better utilization of the available power and optimal use of the energy storage in the cluster as shown in Figure 11b. Furthermore, the battery currents were able to track the reference value very closely. On top of those, the proposed technique enhanced the dynamic response significantly compared to the conventional consensus algorithm. It is noticed that the proposed design achieves fast voltage recovery with less settling time, overshoot/undershoot, and rising time, so the system operation becomes more reliable and steadier under the abovementioned critical operating conditions, which ensures the effectiveness of this proposed technique.

**Author Contributions:** Conceptualization, methodology, writing—original draft preparation, Z.H.A.A.-T.; writing—review and editing, supervision, investigation, and visualization, T.T.L., G.F. and F.B. All authors have read and agreed to the published version of the manuscript.

**Funding:** This research received no external funding.

**Institutional Review Board Statement:** Not applicable.

**Informed Consent Statement:** Not applicable.

**Data Availability Statement:** Not applicable.

**Acknowledgments:** The authors thank Al-Furat Al-Awsat Technical University's financial support, and the research work is also part of the MBIE SSIF AETP entitled Future Architecture of the Network.

**Conflicts of Interest:** The authors declare no conflict of interest.

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
