# Peer review of "Optimal Coordinated Control Strategy of Clustered DC Microgrids under Load-Generation Uncertainties Based on GWO"

_electronics, doi:10.3390/electronics11081244_

Round 1
Reviewer 1 Report
Optimal coordinated control strategy of Clustered DC micro-grids under load-generation uncertainties based on GWO
The paper describes a new control strategy with GWO. Here are some specific comments I would like the authors to address.
Specific Comments:
- The article needs to be checked clearly. Please fix the typos such as (line 197) “I” should change to be” I”, (line 233) “charatersied” should be corrected.
- In figure 9, the output voltage drops deeply in transient state but it may perform good in steady state. Because the waveforms are smooth in figure 9(b),(c) and (e). Compare figure 10(e) with figure9(e), does it mean that the proposed method adjusted the parameters-Kp and Ki too fast?
- Additional waveforms need to be shown in figure 9 and figure 10, voltage regulation, voltage ripple and noise, current ripple.
- In conclusion, the stable is a merit for the proposed method. But the stable need proved seriously, it lacked proof.
- Line218-line221 mentioned about advantages ,these may add to conclusion if simulations can prove these advantages.
- Results and Discussion mentioned the output voltage values, it may be more clear for increasing percentage at overshoot and undershoot.
- Add a completed experiment may increase the article’s integrity.
Date of this review
27th March 2022
Reviewer 2 Report
This paper presents a method to increase the robustness of MGs system. Because of the unexpected generated powers and instability of load, it is meaningful to discuss this problem.
1.The main contribution of this paper uses GWO to optimize PI controller parameters. The authors choose GWO but not other heuristic algorithms such as genetic algorithm without enough reason. Authors should add the corresponding simulation.
2.The contribution was not apparent. The authors just applied the well-known methods to a different system.
3.In Figure 9, all images have the same x-axis, but the image sizes are different. Figure 10 has the same problem.
4.Simulation results from different methods can be put together (such as Figure 9(a) and Figure 10(a)). The current arrangement makes comparisons very difficult.
Reviewer 3 Report
The manuscript entitled “Optimal coordinated control strategy of Clustered DC microgrids under load-generation uncertainties based on GWO” has been investigated in detail. The topic addressed in the manuscript is potentially interesting and the manuscript contains some practical meanings, however, there are some issues which should be addressed by the authors:
- In the first place, I would encourage the authors to extend the abstract more with the key results. As it is, the abstract is a little thin and does not quite convey the interesting results that follow in the main paper. The "Abstract" section can be made much more impressive by highlighting your contributions. The contribution of the study should be explained simply and clearly.
- The readability and presentation of the study should be further improved. The paper suffers from language problems.
- The “Introduction” section needs a major revision in terms of providing more accurate and informative literature review and the pros and cons of the available approaches and how the proposed method is different comparatively. Also, the motivation and contribution should be stated more clearly.
- The importance of the design carried out in this manuscript can be explained better than other important studies published in this field. I recommend the authors to review other recently developed works.
- The performance of the proposed method should be better analyzed, commented and visualized in the experimental section.
- Figure 8 should be improved.
- What makes the proposed method suitable for this unique task? What new development to the proposed method have the authors added (compared to the existing approaches)? These points should be clarified.
- “Results and Discussion” section should be edited in a more highlighting, argumentative way. The author should analysis the reason why the tested results is achieved.
- The authors should clearly emphasize the contribution of the study. Please note that the up-to-date of references will contribute to the up-to-date of your manuscript. The studies named- A new hybrid model for wind speed forecasting combining long short-term memory neural network, decomposition methods and grey wolf optimizer; Grasshopper optimization algorithm-based adaptive control of extruder pendulum system in 3D printer- can be used to explain the proposed method in the study or to indicate the contribution in the “Introduction” section.
- How to set the parameters of proposed method for better performance?
- The complexity of the proposed model and the model parameter uncertainty are not enough mentioned.
- The effect of the parametric uncertainty is not discussed in detail. How did the comparison methods perform with or without the uncertainty?
- The convergence analysis of the proposed method is not given. This analysis should be given.
- It will be helpful to the readers if some discussions about insight of the main results are added as Remarks.
This study may be proposed for publication if it is addressed in the specified problems.
Round 2
Reviewer 1 Report
Thanks for your responses. Each comment has been explained clearly and the article has been increased some contents for improving the readability . There is no question for my opinion.
Reviewer 2 Report
This paper can be accepted.
Reviewer 3 Report
The revised manuscript entitled “Optimal coordinated control strategy of Clustered DC microgrids under load-generation uncertainties based on GWO” has been investigated in detail.
- The “Introduction” section needs a major revision in terms of providing more accurate and informative literature review and the pros and cons of the available approaches and how the proposed method is different comparatively. Also, the motivation and contribution should be stated more clearly.
- The importance of the design carried out in this manuscript can be explained better than other important studies published in this field. I recommend the authors to review other recently developed works.
- Figure 8 should be improved.
- The effect of the parametric uncertainty is not discussed in detail. How did the comparison methods perform with or without the uncertainty?
- The convergence analysis of the proposed method is not given. This analysis should be given.
- The authors should clearly emphasize the contribution of the study. Please note that the up-to-date of references will contribute to the up-to-date of your manuscript. The study named- “A new hybrid model for wind speed forecasting combining long short-term memory neural network, decomposition methods and grey wolf optimizer”- can be used to explain the proposed method in the study or to indicate the contribution in the “Introduction” section.
- How to set the parameters of proposed method for better performance?
- The complexity of the proposed model and the model parameter uncertainty are not enough mentioned.
Round 3
Reviewer 3 Report
It is acceptable in the present form.